# Seed Yield, Crude Protein and Mineral Nutrients of Lentil Genotypes Evaluated across Diverse Environments under Organic and Conventional Farming

**DOI:** 10.3390/plants11233328

**Published:** 2022-12-01

**Authors:** Miltiadis Tziouvalekas, Evangelia Tigka, Anastasia Kargiotidou, Dimitrios Beslemes, Maria Irakli, Chrysanthi Pankou, Parthena Arabatzi, Maria Aggelakoudi, Ioannis Tokatlidis, Athanasios Mavromatis, Ruijun Qin, Christos Noulas, Dimitrios N. Vlachostergios

**Affiliations:** 1Institute of Industrial and Forage Crops, Hellenic Agricultural Organization—DEMETER, 41335 Larissa, Greece; 2ALFA SEEDS SA, 41500 Larissa, Greece; 3Institute of Plant Breeding and Genetic Resources, Hellenic Agricultural Organization—DEMETER, 57001 Thessaloniki, Greece; 4Department of Agricultural Development, Democritus University of Thrace, 68200 Orestiada, Greece; 5Laboratory of Genetics & Plant Breeding, Faculty of Agriculture, Aristotle University of Thessaloniki, 54124 Thessaloniki, Greece; 6Department of Molecular Biology and Genetics, Democritus University of Thrace, 68100 Alexandroupolis, Greece; 7Hermiston Agricultural Research and Extension Center, Oregon State University, Hermiston, OR 97838, USA

**Keywords:** lens culinaris yield, seed quality traits, varieties, year × location, farming system

## Abstract

Lentil is an important legume crop for human and animal dietary needs due to its high nutritional value. The effect of genotype and growing environment was studied on seed yield (SY), crude protein (CP) and mineral nutrients (macro and micronutrients) of five lentil genotypes grown at four diverse locations for two consecutive years under organic and conventional farming. The location within each year was considered as a separate environment (E). Data were subjected to over environment two-way analysis of variance, while a genotype (G) plus genotype × environment (GGE) biplot analysis was performed. Our results indicated the E as the main source of variation (62.3–99.8%) for SY, CP and macronutrients for both farming systems, while for micronutrients it was either the E or the G × E interaction. Different environments were identified as ideal for the parameters studied: E6 (Larissa/Central Greece/2020) produced the higher CP values (organic: 32.0%, conventional: 27.5%) and showed the highest discriminating ability that was attributed to the lowest precipitation during the crucial period of pod filling. E7 (Thessaloniki/Central Macedonia/2020) and E8 (Orestiada/Thrace/2020) had fertile soils and ample soil moisture and were the most discriminating for high micronutrient content under both farming systems. Location Orestiada showed the highest SY for both organic (1.87–2.28 t ha^−1^) and conventional farming (1.56–2.89 t ha^−1^) regardless the year of cultivation and is proposed as an ideal location for lentil cultivation or for breeding for high SY. Genotypes explained a low percentage of the total variability; however, two promising genotypes were identified. Cultivar “Samos” demonstrated a wide adaptation capacity exhibiting stable and high SY under both organic and conventional farming, while the red lentil population “03-24L” showed very high level of seed CP, Fe and Mn contents regardless E or farming system. This genetic material could be further exploited as parental material aiming to develop lentil varieties that could be utilized as “functional” food or consist of a significant feed ingredient.

## 1. Introduction

Increasing crop productivity using sustainable agricultural practices while at the same time maintaining soil health and preserving the quality of the environment are among the most important challenges for modern agriculture in the coming decades. This is in line with the ever-growing world population which by the year 2050 is estimated to reach 9.1 billion [1,2]. Additionally, climate change and intensive agricultural systems constitute the crucial difficulty agriculture and food security confronts in the 21st century together with the demand for adopting environmentally friendlier methods of producing a larger quantity and better quality of food products [3].

Lentil (*Lens culinaris* Medik.) is an important legume for human and animal dietary needs [4,5] as its seeds have recently been classified as functional food, due to the high nutritive value, polyphenols and other bioactive compounds [6,7]. It is widely known that lentil seeds are an abundant source of protein storage, that constitute a viable alternative to animal protein, supplying the human body with essential and non-essential amino acids which are important for a healthy diet [6,7]. The protein concentration in the seeds of a large number of lentil species ranges from 22 to 36% [8]. Several studies indicate that the consumption of lentils is positively correlated with protection against various illnesses such as diabetes, obesity cardiovascular disease and cancers including colon, thyroid, liver, breast and prostate [9,10,11,12,13]. Additionally, lentil seeds contain low amounts of fat and sodium (Na), but high levels of potassium (K) (1:30 Na:K) [14]. Potassium is the most abundant element fluctuating from 7.8 g/kg to 8.6 g/kg in the kernel and 5.4 g/kg to 13.7 g/kg in the whole seed [15]. Several minerals such as iron (Fe), zinc (Zn), copper (Cu), manganese (Mn), molybdenum (Mo), selenium (Se) and boron (B) and vitamins (thiamine, riboflavin, niacin, pantothenic acid, pyridoxine, folate, α, β and γ tocopherols and phylloquinone) have been well documented in lentils [16,17,18]. A percentage of 40.5 to 42.9% of the complete phosphorus (P) of entire lentils is contained in phytic acid. A higher percentage of around 43.7–44.0%, of the whole P is contained in phytic acid in the kernel [15]. The content of Zn ranges between 3.2 mg/100 g and 6.3 mg/100 g, whereas significant quantities of Fe are also present in lentil seeds [6]. The consumption of lentil in the daily diet prevents Fe deficiency anemia, especially in children and women [19] and amounts of Fe are in highly bioavailable form and more digestible when lentils are cooked, germinated or fermented [20]. Zinc deficiency is related with weakened growth and development, compromised immune system, cardiovascular diseases and chronic kidney illnesses in humans [21]. The results of Khazaei et al.’s [7] study confirmed the high quantity of seed Fe and Zn in lentils, and they found molecular markers (Single Nucleotide Polymorphisms SNPs) that could be used for marker-assisted selection to improve Fe and Zn concentration in lentil seeds in sufficient qualities.

Nowadays, consumers are re-evaluating their diets in favor of a balanced diet for a healthy way of life. Therefore, lentil production in the world is increasing annually, reaching 6.5 million tons in 2020, according to FAO data (Food and Agriculture Organization of the United Nations, FAOSTAT (http://www.fao.org/faostat/en/#data/QC accessed on 20 January 2021). Lentil is a well-adapted plant that grows in a wide range of climate and soil conditions. It is cultivated in Mediterranean and subtropical dryland regions, and usually no synthetic fertilizers are applied for cultivation due to the ability to fix atmospheric nitrogen (N_2_). Therefore, lentils can be very well integrated into organic and conventional crop rotations. In Greece, lentils are cultivated in non-irrigated fields and the annual production is about 13,300 tons from an area of about 11,000 ha [22].

In order to achieve global food security, lentil breeders face a major challenge which is to simultaneously increase both yield and protein content. Many researchers reported negative correlation between seed protein and seed yield and also high heritability (0.84) for protein content [23,24,25]. The differences in both agronomic and seed properties in lentils is the result of genetic and environmental factors [24,26,27,28,29]. The seed yield is ultimately the result of interaction of genotype with the environment. Stable performance over different environments is an advantageous trait, which depends on the extent of G × E interactions [30]. Vlachostergios et al. [31] found that environments influence seed yield, agronomic traits, protein content and cooking time of lentil varieties evaluated in many regions in Greece where lentils are cultivated. Karagounis et al. [32] analyzed three cultivars under organic and conventional farming system and detected that the concentration of physicochemical traits and chemical analysis (P, K, Ca, Mg, Zn and Fe) in seeds are significantly influenced by the environment. In another study, 36 lentil cultivars were evaluated in an organic and conventional agriculture system for three years and high diversification was identified in physicochemical properties [33]. Further research underlined that the genotype and the growing environment affect the phytochemical content and the antioxidant activities from lentil cultivars evaluated in multi-location trials [34]. Ansari & Jha [35] deploy modern “omic” technologies aiming to exploit the genetic variability for micronutrient and protein content in seeds to increase the nutritional value of legumes. They concluded that the concentration of these nutrients is remarkably influenced by soil composition and environmental factors.

The objectives of this study were: (i) to investigate the environmental and genotypic effect on seed yield (SY), crude protein (CP) and mineral nutrients (macro and micronutrients) of five lentil genotypes grown at four diverse locations under two farming systems, (conventional and organic), for two consecutive years; and (ii) to identify superior genotypes with elevated values of the measured parameters which could be further exploited as parental material in breeding programs and suggested for as “functional” food to cover certain deficiencies in human diets or consist a significant animal feed ingredient.

## 2. Results

### 2.1. Combined Variance Analysis for Yield and Inorganic Compounds of Lentil Genotypes

Genotype (G), environment (E) and their interactions (G × E) were found to be a significant (*p* ≤ 0.01, *p* ≤ 0.001) source of variation for all examined traits (SY, CP, P, K, Fe, Cu, Mn and Zn) on both organic and conventional farming (Table 1). In organic farming, except for Fe and Mn, the other traits showed a considerable extent of variation due to E that ranged from 62.3 to 97.32%. In the case of seed Fe and Mn, the greatest proportion of variation was attributed to G × E interaction at 56.5% and 67.1%, respectively. Similarly, in conventional farming, E was the significant source of variation for SY (80.3%), seed CP (94.5%), seed P (92.8%), K (99.8%), Fe (53.2%) and Cu (58%), whereas for Mn (50.3%) and Zn (47.5%) G × E interaction contributed to the highest portion of total variance. Oppositely lower variation in all parameters was found due to G which varied from 0.8% to 7.7% in organic and from 0.1% to 17.6% in conventional farming.

According to the means comparison presented in Table 2, significant differences (*p* < 0.05) were detected for SY, seed CP, seed macronutrients (P, K) and micronutrients (Fe, Cu, Mn, Zn), among the five lentils genotypes averaged across eight growing environments on both farming systems. Specifically, in organic farming, SY ranged from 1.27 t ha^−1^ (G5) to 1.41 t ha^−1^ (G1) and the highest SY levels were obtained by G1 along with G4, whereas the lowest value was monitored in the red lentil population “03-24L” (G5). Seed CP and P varied significantly among the tested genotypes, ranging from 23.3% (G1) to 25.8% (G4) and 0.34% (G5) to 0.37% (G1), respectively. Among the genotypes, only G5 showed considerably lower seed K concentration (0.9%); however, the fluctuation among genotypes was small. Red lentil “03-24L” (G5) recorded a remarkable increase in seed Fe content, reaching values of 177.7 mg kg^−1^, whereas cultivar ‘Dimitra’ (G2) recorded values at approximately 103.9 mg kg^−1^. For the rest of the seed micronutrient contents (i.e., Cu, Mn and Zn), significant differences (*p* < 0.05) were also recorded among genotypes, whereas G5 had the lowest content of Cu (8.2 mg kg^−1^) and Zn (47.3 mg kg^−1^) and the highest content of Mn (18.2 mg kg^−1^).

Similarly, in conventional farming, significant differences (*p* < 0.05) were detected among the five lentils genotypes for all examined traits as presented in Table 2. Genotype ‘Elpida’ (G3) had the lowest SY (1.36 t ha^−1^), whereas genotype ‘Samos’ (G1) again showed the highest SY. In contrast, G1 had the lowest concentration in seed CP, P and K, whereas genotype “Elpida” (G3) had the highest seed P and K concentration and red lentil population “03-24L” (G5) the highest seed CP concentration. Although a high content of seed Fe was recorded in all studied genotypes, it was evident that seed Fe content of the red lentil population “03-24L” (G5) was approximately 50% higher compared to other genotypes, as was also observed in organic farming.

Additionally, significant differences among genotypes were found in the contents of seed Cu, Mn and Zn, averaged across environments, with cultivar “Dimitra” (G2) reaching the highest content estimated at 21.0 mg kg^−1^ and 51.5 mg kg^−1^ for Mn and Zn, respectively, and 9.3 mg kg^−1^ for Cu along with cultivar “Elpida” at 9.5 mg kg^−1^.

Significant differences (*p* < 0.05) were detected among the eight growing environments for SY, seed CP, seed macronutrient (P, K) and micronutrient (Fe, Cu, Mn, Zn) contents, averaged across genotypes, both in organic and conventional farming (Table 3). In organic farming, lentil grown in Larissa 2020 (E6) recorded the lowest values (viz. 0.46 t ha^−1^) in SY, while lentil grown in Orestiada the same year (E8) recorded the highest values in SY (viz. 2.28 tha^−1^). Likewise, seed CP significantly varied across environments where Orestiada 2019 (E4) and Domokos 2019 (E1) recorded the lowest values and Larisa 2020 (E6) the highest values. There were also significant differences (*p* < 0.05) in seed macronutrient contents originated from the eight environments. Orestiada 2020 (E8) produced lentils with the lowest seed P and K content, while Domokos 2020 (E5) and Thessaloniki 2020 (E7) produced lentils with the highest seed P and K content. Additionally, significant environmental variation existed for seed micronutrient contents (Fe, Cu, Mn and Zn). It was noticed that among the tested environments, the highest values of seed Fe (266.9 mg kg^−1^) were recorded in Thessaloniki 2020 (E7) and the lowest Fe content (71.9 mg kg^−1^) in Larissa 2019 (E2). Among the eight environments Larissa in 2020 (E6) produced lentil seeds with the highest Cu concentration and Orestiada in 2019 (E4) with the highest seed Mn and Zn concentration, while the lowest values for Cu and Zn were recorded in Orestiada 2020 (E8) and for Mn in Domokos 2020 (E5). The rest of the environments gave moderate values in lentil seed micronutrient.

Similarly, significant differences were observed in conventional farming for SY, CP, P, K, Fe, Cu, Mn and Zn among all the environments tested (Table 3). The highest SY was recorded in the second growing season in Orestiada (E8), while lentil grown in Larissa 2019 (E2) exhibited the lowest SY followed by Domokos 2020 (E5) and Thessaloniki 2020 (E7). There were also significant differences (*p* < 0.05) in seed P, K contents and seed CP concentration originated from the eight environments. Larissa 2019 (E2) produced lentils with the lowest content on both CP and K, while Orestiada 2020 (E8) produced lentils with the lowest content in seed P. The rest of the environments gave moderate values in these measurements. Significant environmental variation existed for seed micronutrient content. Lentil grown in Thessaloniki 2020 (Ε7) recorded the highest seed Fe, Cu and Mn concentration and those grown in Orestiada 2019 (E4) the highest Zn concentration.

### 2.2. GGE Biplot Analysis for Seed Yield in Organic and Conventional Farming

Figure 1 illustrates a biplot analysis of SY of five lentil genotypes across eight environments in organic farming system. Biplot analysis explained 91.7% of the total G plus G × GE variation. The “which-won-where” view of the GGE biplot (Figure 1a) consists of an irregular polygon and a set of lines drawn from the biplot origin and intersecting each of the sides at right angles. This polygon view separated the eight environments into three sectors with different winning genotypes. Analytically, G2 and G4 were the highest yielding genotypes in environments E5, E8 and E4, G1 the highest yielding genotype in environment E7 and slightly in environments E1 and E2 and G5 in environments E3 and E6. These results suggest that the target environment was divided into three mega-environments. In Figure 1b, the single-arrowed line, the “Average Environment Coordination” (“AEC”) abscissa, points to a higher mean yield across environments. Hence, genotypes G4, G2 and G1 produced the highest SY. Additionally, the double-arrowed line, the ordinate, points to greater variability (poorer stability) in either direction; therefore, the more stable genotype is G5, followed by G3 and G4. Thus, the winning genotype is G4 characterized by both high yield and stability.

Biplot analysis of SY of five lentil genotypes across the eight environments in conventional farming system is illustrated in Figure 2a,b. The biplot analysis explained 85.9% of the total variability. The polygon view in Figure 2a showed that the eight environments fell into three main sectors with different winning genotypes. Specifically, the first sub-environment included the environments E5 and E3 that coincide with E6 and revealed that G2 was the highest yielding genotype. In the second sub-environment that includes E1, E2 and E4, the best performing genotype is G1 and the third sub-environment formed by environments E7 and E8 in which G4 is the winning genotype. In Figure 2b, according to the single-arrowed line, the “Average Environment Coordination” (“AEC”) abscissa, G3 and G1 had the highest mean SY, and according to double-arrowed line, the “AEC” ordinate, highly unstable are the genotypes G4, G2, G5 and G3. Overall, G1 can be characterized as an “ideal” genotype for seed production and stability in conventional farming, across the eight environments.

### 2.3. GGE Biplot Analysis for Seed Crude Protein and Macronutrient in Organic and Conventional Farming

Figure 3a,b displays the GGE biplot graphical analysis of seed P and K macronutrient contents and % seed CP of five lentil genotypes in the eight environments cultivated in organic farming system. Biplot analysis explained 87.8% of total variability in Figure 3a, and 89.6% in Figure 3b. In the “which-won-where” view of Figure 3a, genotypes G4, G5, G3 and G1 are located on the vertices of the polygon performed either the best or the poorest in one or more measurements, i.e., % CP, seed P and seed K contents. The biplot is divided into sectors by the equality lines (sectors of convex hull). Consequently, genotype G4 became the most promising genotype in terms of seed CP since is located on the respective vertex. Similarly, genotype G1 is the most promising in terms of seed P and seed K content since is located on the vertices of the polygon in the sector where seed P and seed K contents are placed. In Figure 3b, the “Average-Environment Axis” (AEA) passes through the average environment and the biplot origin; therefore, a test environment that has a smaller angle with the AEA is more representative of other test environments. Thus, environment E6 is uniquely correlated with seed % CP, and seed K and P contents. Additionally, broad correlated environments with the traits under study were environments E7, E8 and E5, while E1, E2, E3 and E4 presented a low correlated environment. Additionally, environments E8, E6, E5 and E1 were the most discriminating.

The GGE biplots in Figure 4a,b present the seed CP and seed macronutrient contents (P, K) in conventional farming system. The “which-won-where” pattern revealed 94.3% of total variation, while in Figure 3b the “Discriminating power vs. representativeness” pattern revealed 98.1% of total variation. Figure 4a, was separated into three sectors (sectors of convex hull) with a different winning genotype in each sector. Therefore, G3, located in the upper right vertical, is the winner genotype for the seed macronutrients. In the same manner, genotype G5 is uniquely correlated with the seed CP. Additionally, genotypes G1 and G4 performed worst for the measurements under study. According to the trait by environment biplot, averaged across genotypes (Figure 4b), the most representative environment for seed P, K contents and seed CP is E5 since it formed the smaller angle with the “Average-Environment Axis” (AEA, or average-tester-axis). Next representative environments are E6 and E7. Furthermore, according to the concentric circles on the biplot, the most discriminating environment is E6 followed, once again, by E7 and E5. The GGE biplot analysis in Figure 5a,b illustrates data for seed micronutrient contents, i.e., Fe, Cu, Mn and Zn of five lentil genotypes across the eight environments in organic farming system. The first two principal components of GGE biplot analysis in Figure 5a explained 97.9% of total variation and in Figure 5b explained 78.2%.

### 2.4. GGE Biplot Analysis for Seed Micronutrient in Organic and Conventional Farming

The GGE biplot analysis in Figure 5a,b illustrates data for seed micronutrient contents, i.e., Fe, Cu, Mn and Zn of five lentil genotypes across the eight environments in organic farming system. The first two principal components of GGE biplot analysis in Figure 5a explained 97.9% of total variation and in Figure 5b explained 78.2%.

According to the polygon view of the “which win where” in Figure 5a, genotype G5 performed highly for seed Fe and Mn content and therefore can be characterized as the winning genotype for these two seed micronutrients. Likewise, G2 is the winning genotype for seed micronutrient Zn and Cu content. The concentric circles view of “Discriminating power vs. representativeness” in Figure 5b indicated that environment E8 was the most discriminating followed closely by environment E7, while E6 and E1 were the least discriminating for seed Fe, Cu, Mn and Zn content. Furthermore, the most representative environments for the measurements under study were E6 with an almost zero-degree angle with the “Average-Environment Axis” (AEA). Additionally, E4 and E7 were sufficiently representative since their vectors formed a small angle with the “Average-Environment Axis” (AEA).

The GGE biplot graphical analysis for seed micronutrient contents (Fe, Cu Mn and Zn) in conventional farming system is presented in Figure 6a,b. In Figure 6a, the first and second principal components (PC1 and PC2) together can explain 96.7% of the total variation and in Figure 6b the 93.5%.

From the polygon view of the biplot analysis in Figure 6a, the genotypes fell into four sections. G5 is the winner genotype in the upper section where seed Fe content is located. G2 is the winner genotype in the right section where seed Cu, Zn and Mn content are located. Genotypes G4, G1 and G3 are relatively the least promising genotypes for seed micronutrient content. The “ideal test environment” is defined as the environment that is most discriminating and also representative among all test environments. Consequently, from the concentric circles view of Figure 6b, E7 and E8 were the most discriminating environment for producing lentil seeds with high micronutrients content, since they exhibit the longest vectors. Additionally, the most representative environments for the seed Fe, Cu, Mn and Zn were again environments E7 and E8 since these have the smaller angle with the “Average-Environment Axis” (AEA), highlighting them as “ideal environments”.

## 3. Discussion

### 3.1. Seed Yield

The highest percentage of the total variability for lentil SY was explained by the environment (E = location × year) followed by the G × E interaction, for both organic and conventional farming system (Table 1). The strong effect of the environment on legume species yield is considered as one of the main reasons for the small expansion of legume cultivation in comparison with cereals [36,37,38,39]. Especially for lentils, high environmental effect on SY have been previously reported by Vlachostergios et al. [21,33] and Khazaei et al. [40], whereas Dehghani et al. [41] who studied the stability of 11 different lentil genotypes in 20 rain-fed environments in Iran found more counterbalanced environment (51.5%) and G × E interaction (45.9%) effects on SY.

This study was not designed to include the farming system as a source of variation; however, our results demonstrated that SY was affected by the farming system. These results were also confirmed in previous studies [42,43,44] although genotypic response within farming system was not thoroughly studied. Under organic farming conditions and across the five lentil genotypes, the most productive environment was E8 (2.28 t ha^−1^) which corresponded to location Orestiada/2020, followed by E3 (1.88 t ha^−1^) (Thessaloniki/2019) and E4 (1.87 t ha^−1^) (Orestiada/2019). Under conventional cultivation the most productive environment was again E8 (2.89 t ha^−1^), followed by E1 (1.66 t ha^−1^) (Domokos/2019) and E4 (1.56 t ha^−1^) (Table 3). It is apparent that in Orestiada, lentil genotypes showed the highest SY regardless of the year of cultivation or the farming system, and therefore, Orestiada could be proposed as the optimum location for lentil cultivation. This could be attributed to the special climatic conditions prevailing in Orestiada (adequate humidity in the critical period of anthesis to pod filling for both seasons) and to favorable soil conditions. The soils in the location of Orestiada where lentil was grown belonged to the order of *Fluvisols* (as also in other two of the four study locations), which are broadly considered very fertile soils, typically with high levels of clay and soil organic matter (SOM) (i.e., in Orestiada SOM was at medium levels as compared to other locations). Moreover, at both E4 and E8, the period which represents the beginning of anthesis to pod filling amounted to high precipitations and favorable temperatures (Table 4). The seedling and flowering stages are the most sensitive to water availability and drought stress, [45] and as reported by Hamdi et al. [46] a deficiency of water during any growth stage in lentil, as in any legume species, often results in a loss of SY. Lentil SY was also the lowest in a dry year (1.47 t ha^−1^) as compared to the highest (2.22 t ha^−1^) in a year with season precipitation was close to the normal across four lentil cultivars and five locations in Montana, USA [47].

GGE biplot analysis revealed three mega-environments for organic and three for conventional farming system (Figure 1a and Figure 2a). It is noteworthy that different combinations of sub-environments consisted of each mega-environment for each farming system, a fact that further highlights Environment as a crucial parameter that determines SY. Furthermore, GGE bi-plot analysis showed a more specific connection between mega-environments and genotypes. Thus, under organic farming, G2 and G4 were the most productive genotypes for E4, E5 and E8, G1 was the best for E7, E1 and E2, while G5 and G3 were the best for E3 and E6. On the other hand, under conventional farming, G1 was the most productive for E1, E2 and E4, G2 for E5, E3 and E6, while G4 was the best for E7 and E8.

Genotypic contribution on the total variation for SY in the study was very low; however, results revealed significant differences between genotypes for both farming systems (Table 1). G1 ranked at the top under organic and conventional farming with a SY performance of 1.41 t ha^−1^ and 1.67 t ha^−1^, respectively (Table 2). According to GGE biplot analysis when genotypes were evaluated for their mean performance and stability, G4 was the more stable and productive genotype under organic farming, followed by G1 (Figure 1b). Under conventional farming G1 was the ideal entry showing high yield performance and stability (Figure 2b). Consequently, G1 (clv ‘Samos’) could be proposed as the ideal genotype for conventional farming, while for organic farming, G1 along with G4 (clv. ‘Thessalia’) were the most adaptable genotypes, according to Yan and Kang [50] who defined an ideal genotype as being the highest yielding genotype across test environments with a stable performance.

### 3.2. Seed Crude Protein

Seed crude protein (CP) protein is an important nutritional trait for food and feed industry and in the present study the largest proportions of its total variance was explained by environmental factors (pedoclimatic conditions) under both farming systems (organic: EV% = 90.1% and conventional: EV% = 94.5%). Consequently, lower proportions of total variance were explained by the G × E interaction (5.8% and 4.1%, respectively) and the genotypes (Table 1). However, G and G × E effects were highly significant for CP indicating that this seed quality trait can be genetically modified but can also be influenced by the environment (pedoclimatic fluctuations). Several other studies have also reported genetic and environmental effects that impact on lentil protein [24,47,51,52]. Little G × E interaction indicates that plant breeders may develop for growers a lentil genotype with high seed protein enriching ability to consistently produce high seed protein-concentration across geographical locations in Greece. Nevertheless, the relatively narrow set of pedoclimatic conditions included in this study and the consequent low G × E interaction reported, may indicate that it is possible that some of the genotypes, especially the ones showing low ability to exploit the high-yielding environments (G5 and G2 under organic and G3 and G5 under conventional), might show better adaptation to other environments. Such differences in genotype adaptation could also be part of the explanation of the negative yield–protein correlation usually observed. Genotype ‘Samos” (G1), the highest yielding genotype under both farming systems, performed worst and showed the lowest CP (Table 2; Figure 4a), whereas G5, a low yielding genotype (with red cotyledons), is the winner for producing the highest CP (Figure 4a) confirming the negative yield–protein correlation usually observed for these two traits [23,24,53]. With respect to environments the highest CP was recorded in E6 the environment with the highest discriminating ability (Figure 4b) probably because of the lowest precipitation during the crucial period of pod filling (April–May) in both years (Table 4) which negatively impaired the starch synthesis, resulting in an increase of seed CP content. However, further studies are needed to explore the underlying mechanisms of the environmental impact on CP content.

### 3.3. Seed Macronutrients

Seed mineral macronutrient contents (P, K) were also influenced by the environment (EV% > 92%) with the exception of seed K under organic farming where G × E effects explained 32.4% of the total variation (Table 1). These results are partially in agreement with those of Chen et al. [47] who found that beside environmental effects, cultivar selection is also very influential on lentil seed mineral nutrient concentrations (including P, K and secondary macronutrients, S, Ca, Mg) for lentils grown after wheat, barley, alfalfa or chemical fallow. On the other hand, our results comply to those by Vandemark et al. [18] who found that for the majority of minerals in lentil seed the highest interaction effect was the location × year (i.e., Environment) effect in two locations in Washington (US). Other authors have reported significant genotype effects on seed macro and microelement concentrations in a large set (46) of lentil genotypes in Eastern Turkey [54] and in 35 advanced breeding lines in Saudi Arabia [55] but Ray et al. [56] reported that despite significant genotypic variation, no significant year × location effects were found on lentil seed K content, even though their lentil data were derived from only two locations in Saskatchewan, Canada, which may have caused the smaller contribution of location relative to other factors.

Among genotypes and across all environments, G1 (‘Samos’) exhibited the highest seed P content (0.37%) under organic farming but the lowest P content (0.34%) under conventional farming system. The least seed K accumulation was also recorded for the same genotype (G1) under conventional farming system (0.92%) (Table 2). Despite significant differences among genotypes for seed P and K contents within each farming system, values either within or between the two systems are quite close for each one of P or K. These results tend to agree with those by Karagounis et al. [32] who found that the same genotype (‘Samos’) showed no significant differences in the percentage of P, K, (and Ca, Mg) between organic and conventional farming. According to “which-won-where” view of GGE biplot analyses, genotype G1 was the most promising in terms of seed P and seed K content under organic farming (Figure 3a), whereas in conventional farming, G1 and G4 performed worst for these two traits (Figure 4a).

Across all genotypes, seed P ranged from 0.20% in E8 (Orestiada/2020) to 0.49% in E6 (Larissa/2020) and the ranking of the environments was similar in the two cropping systems (organic and conventional) (Table 3). For seed K, contents ranged from 0.80% in E8 to 1.30% in E7 (Thessaloniki/2020) under organic farming and from 0.87% in E2 (Larissa/2019) to 1.20% in E5 (Domokos/2020) under conventional farming. On absolute terms, seed P and K contents are more or less close to or slightly different from the ranges (P: 0.29–0.41%; K: 0.81–0.92%) reported in the study by Chen et al. [47] for 15 environments in Montana (US) and comparable to those (P: 0.40–0.46%; K: 0.96–1.06%) reported in the study by Vandemark et al. [18] in two locations in Washington (US). Notably, seed P and K contents were higher in the second wetter year (2020) as compared to those in 2019 growing season under both farming systems indicating that ample soil moisture conditions may have favored the uptake and accumulation of these nutrients from soil and translocation to the seed. These results confirm previous findings in that seed mineral concentration of pulse crops can be affected among other variables (harvest time, intercropping, mycorrhizal colonization, soil nutrient availability) by soil moisture [56,57].

### 3.4. Micronutrients

Environment (Y × L) provided again the greatest component of variance for most of the seed micronutrients and more specifically for the minerals Fe (conventional farming), Cu (conventional and organic), Zn (organic). Conversely, highly significant and high were the G × E effects for Fe (organic farming), Mn (conventional and organic) and Zn (conventional) (Table 1). Genotypes again explained the least of the total variance for all micronutrients. These results are consistent with those of Vandemark et al. [18] who studied mineral concentrations of lentil cultivars and breeding lines grown in the US Pacific Northwest and indicate that the low genotype effects may require a more diverse set of genotypes to identify useful genetic variation for these microelements.

Based on the polygon view of the “which win where” GGE biplot analyses genotype G5 (‘03-24L’) performed the best for seed Fe under both farming systems and for seed Mn under organic farming (Figure 5a and Figure 6a; Table 2). This finding suggests that this population can provide genetic material to develop new cultivars with higher Fe concentrations, an approach that has been proposed as a means of increasing Fe uptake in human diets [18]. Furthermore, the red lentil population ‘03-24L’ has been characterized as a promising genetic material due to its high phenolic contents and antioxidant capacity values across environments [34]. Levels of seed Fe among genotypes and across all environments ranged from 103.9 mg kg^−1^ (G2) to 177.7 mg kg^−1^ (G5) under organic farming and from 114.9 mg kg^−1^ (G4) to 256.5 mg kg^−1^ (G5) under conventional farming (Table 2). Values for seed Fe are considerably high (almost double) of those (54.1–62.1 mg kg^−1^) reported by Chen et al. [47] for 15 environments in Montana (US) and by far higher of those (26–92 mg kg^−1^) reported by Gupta et al. [58] but for lentils grown in a greenhouse experiment. The levels of Cu in the seeds are broadly similar but our results are relatively higher for seed Mn, and Zn as compared to the results by Chen et al. [47]. The range of seed micronutrients of the genotypes tested in our study are consistent with results for 18 lentil cultivars grown in two locations in Saskatchewan, Canada [56] where concentrations ranged for Fe, 75.6 to 100 mg kg^−1^; for Cu, 7.0 to 9.2 mg kg^−1^, for Mn, 12.2 to 14.8 mg kg^−1^ and for Zn, 36.7 to 50.6 mg kg^−1^.

Environments E7 (Thessaloniki/2020) and E8 (Orestiada/2020) were the most discriminating environments for producing lentil seeds with high micronutrient contents under both farming systems (Figure 5b and Figure 6a; Table 3). These environments include broadly fertile soils (*Fluvisols*) and the ample soil moisture recorded in the second growing seasons (especially the critical period from anthesis to pod filling) may have favored the ability of the genotypes to uptake more micronutrients in the seed [57] and could be proposed as ideal for breeding or screening lentil genotypes for micronutrient content.

An improved profile of mineral micronutrients (especially Fe) in lentil genotypes of our study can provide the genetic pool to develop new cultivars with even higher concentrations that would be beneficial for human diets and feed. However, the higher magnitude of environmental effects compared to genotype effects indicates that limited potential exists for increases in seed mineral concentrations through selections among these lentil genotypes. Genotypes that are more efficient at obtaining minerals from growing environments will be more useful as parental materials to develop lentil cultivars with high seed mineral concentrations. Genetic biofortification which is the identification and transfer of the corresponding genes to important crops that increase their uptake capacity, is expected to be the most cost-efficient approach, along with the use of mineral fertilizers to improve mineral nutrient content in diets [59]. Biofortification of minerals in lentil will have a positive impact on maternal and child health in mineral deficiency affected areas. Therefore, identifying lentil genotypes that could produce seeds with high micronutrient density could promote the adoption of lentil crop for biofortification and reduce malnutrition in developing countries where normally have less access to fertilizer inputs [54,60]

## 4. Materials and Methods

### 4.1. Genetic Material

The genetic material consisted of five lentil genotypes. Four commercial cultivars (G1: ‘Samos’; G2: ‘Dimitra’; G3: ‘Elpida’; and G4: ‘Thessalia’) developed by the Institute of Industrial and Forage Crops (IIFC) in Larissa, Greece, and one population (G5: 03-24L) with red cotyledons which was originated from ICARDA and improved by IIFC following the method described by Tokaltidis and Vlachostergios [61]. These genotypes are popular among growers and were selected because of their high yield potential and variation in reaching flowering and maturity earliness. Other basic characteristics of the genotypes can be found in Vlachostergios et al. [31].

### 4.2. Locations and Growing Conditions

The genotypes were evaluated for two consecutive growing seasons (2018–2019 and 2019–2020, hereafter mentioned as 2019 and 2020, respectively) under organic and conventional farming conditions at four locations across Central and North Greece: (1) Thessaloniki in Central Macedonia, North Greece (Latitude 40°32′69″ N, Longitude 22°59′83″ E, elevation 5 m a.s.l.), (2) Orestiada in Thrace, North Greece (Latitude 41°30′14” N, Longitude 26°32′99″ E elevation 26 m a.s.l.), (3) the central farm of IIFC in Larissa, Thessaly, central Greece (Latitude 39°36′81″ N, Longitude 22°25′94″ E, elevation 77 m a.s.l.), and (4) Domokos in Central Greece (or Sterea Hellas) (Latitude 39°1′13″ N, Longitude 22°19′74″ E, elevation 500 m a.s.l.). In this study location and growing season (2019 or 2020), combinations were appended as environments (E). Therefore, environment 1 (E1) is referred to Domokos in 2019, environment 2 (E2) to Larissa in 2019, environment 3 (E3) to Thessaloniki in 2019, environment 4 (E4) to Orestiada in 2019, environment 5 (E5) to Domokos in 2020, environment 6 (E6) to Larissa in 2020, environment 7 (E7) to Thessaloniki in 2020, and environment 8 (E8) to Orestiada in 2020.

The study locations included, basically, two types of climates according to the Köppen–Geiger classification [62,63]. Thessaloniki features a humid subtropical climate (*Cfa*), whereas Larissa, Domokos and Orestiada exhibit a hot-summer Mediterranean climate (*Csa*). Meteorological data including precipitation and daily mean air temperature during the growing season (November to July) were recorded by wireless automatic weather station in each location and trial. The automatic weather station consisted of a data acquisition system and a set of sensors for the measurement of the above-mentioned variables. Meteorological data and basic soil properties are compiled in Table 4.

The testing environments were rather different in terms of temperature and precipitation. E4 (Orestiada in 2019) had the lowest season precipitation and E5 (Domokos in 2020) the highest. Location Domokos either in 2019 (E1) or in 2020 (E5) showed the lowest average temperatures and the highest precipitations in the growing season among the other environments. The highest season average temperatures were recorded in environments E2, E6 and E3, E7 (representing Larissa and Thessaloniki, respectively) which were, however, quite close to the long-run averages (30-year averages, data not shown). Precipitation between April and May (early reproductive till pod filling growth stages) was lower in 2019 and ranged from 72.0 mm (E2) to 118.3 mm (E4) in comparison to the wetter 2020 growing season that ranged from 99.0 mm (E6) to 186.4 mm (E5) (Table 4).

Soil orders included *Fluvisols* and *Vertisols* [48,49]. Briefly, *Fluvisols* are mainly young azonal soils, in alluvial deposits (floodplain), lacustrine (lake) and marine deposits. These soils exhibit very little profile development, evidence of stratification and/or an irregular organic matter profile. *Vertisols* are productive soils if properly managed but become very hard in the dry season and sticky in the wet season.

Soil samples were taken separately from conventional and organic farming treated plots. Topsoil (0–30 cm) physicochemical parameters included the soil texture [64], pH (1:1) [65], soil electrical conductivity (EC, 25 °C) [66], percentage soil organic matter (SOM) [67], calcium carbonate equivalent (%, CaCO_3_) [68] and available phosphorous (P_Olsen_) [69] (Table 4). As can be seen from Table 4, no major differences were found between conventional and organic treated plots with respect to the basic soil properties except of P_Olsen_ in E1, E3, E5 and E6. Most soils had clayey soil texture, slightly alkaline soil pH, and low EC (<1.00 mS cm^−1^). SOM content was mostly low (1.0–1.7%) except for E4 and E8 in organic farming which was medium in SOM content. Soils were either deficient (<10.0 mg P kg^−1^ soil) or moderately sufficient (>10 and <20 mg kg^−1^ soil) in soil P_Olsen_ except of soils in E5 (conventional and organic) and in E1 in organic farming which had P_Olsen_ high (>25 mg kg^−1^ soil).

In all trials the genotypes were arranged in plots following a randomized complete blocks (RCB) design with four replications. The plots consisted of seven (7) plant rows of four (4) m long each with a distance of 0.25 m between the plant rows. The seeding rate was 80–120 kg ha^−1^ and sowing was continuous in the furrows with normal and uniform distribution of the seed. Sowing was performed during the last week of November 2018 and November 2019 in all locations and both farming systems.

The trials at the locations were accounted as rain-fed environments. For the conventional farming practice, a balanced basal fertilization with 160 kg ha^−1^ (0N-46P_2_O_5_-0K_2_O) was incorporated pre-sowing. In the organic farming system, no synthetic fertilizer was applied during the experiments and for at least the last 3 years prior to the experiment. To control weeds, weeding was done at regular intervals by hand, both in conventional and organic conditions. Other phytosanitary practices to control diseases and pests were followed according to the local recommendations in conventional system. In the organic farming system, no phytosanitary practice was performed.

### 4.3. Data Sampling, Sample Preparation and Measurements

#### 4.3.1. Seed Yield and Crude Protein

When each genotype reached its physiological maturity (over 50% of the plants of each plot), plants were hand-harvested using hand sickles to ground level and threshed using a laboratory thresher (Wintersteiger LD350) in order to assess seed yield (SY). SY was calculated on a plot basis (3m^2^ per plot) and converted to t ha^−1^ after adjusting to 13% seed moisture content. For the assessment of the seed crude protein concentration (% CP) at physiological maturity, samples consisting of the 3 inner plant rows of the plots were cut, bagged and dried to a constant weight in a dryer room. These samples were threshed and seed crude protein concentration (% CP) was determined on uniformly ground samples obtained from all plots for each genotype using the Kjeldahl method and expressed as the % CP = total N × 6.25 [70]. The determination of the protein content was carried out in triplicate samples of 0.5 g. The data presented are expressed on a 0% moisture basis.

#### 4.3.2. Seed Micro and Macronutrients

The seed samples were carefully washed with distilled water to remove surface contaminants, such as dust, and unwanted material, such as soil, shoot remains, etc. The samples were placed in labeled porcelain crucibles and oven-dried (memmert UL80, memmert GmbH +Co.KG, Schwabach, Germany) at 70 °C for at least 48 h. A portion of clean seeds were ground using a laboratory mill (Retsch SK 100, Retsch GmbH, Haan, Germany), homogenized into a fine powder and stored in labeled polyethylene bags.

Triplicate dried and homogenized samples of known weight (1.0 g) were digested by a dry-ashing method in a muffle furnace at 450 ± 5 °C for 4 h [71,72]. After cooling, in order to avoid sample losses, 1 mL of distilled water was added in the crucibles to wet the ash. Then, 20 mL of HCl 1M was added to dilute the ash followed by heating at about 80 °C in a hot plate till almost dryness. Following this step HCl 1M was added again and heat for 15 min. The aliquot was collected by filtration through a Whatman No 42 or equivalent filter paper into 50 mL volumetric flasks. The filter paper and the crucibles were washed by distilled water into the volumetric flasks several times to ensure quantitative content transfer and finally the flasks were filled up to the mark with distilled water. A blank sample was also prepared for every batch of samples using the same method and manner in order to identify the potential effects of the whole process on the measured parameters.

In the seed digest total concentrations of iron (Fe), copper (Cu), manganese (Mn) and zinc (Zn) were measured by atomic absorption spectrometry (AAS) using a Thermo iCE 3000 Series instrument with a flame technique (Thermo Fischer Scientific Inc., Waltham, MA, USA) and a gas air-acetylene mixture, connected to an autosampler CETAC ASX-520 (ATS Scientific Inc., Mainway, Burlington, Canada). These were reported as the most commonly used techniques for qualitative and quantitative determination of minerals in food samples [73], because of its specificity, sensitivity, high precision, simplicity, rapid analysis, low cost, low detection limit and wide linear range [74]. For the measurements, mixed standard solutions of the four metals (Fe, Cu, Mn and Zn) were used with concentrations from 0.1 to 10 mg/L using standard stock solutions (1000 mg/L on each of them. These solutions were used to build the calibration curves. The cathode lamp wavelengths for each individual metal were: Cu: 324.8 (±0.5) nm; Fe: 248.3 (±0.2) nm, Mn: 279.5 (±0.2) nm and Zn: 213.9 (±0.2) nm. For all four metals, there was a correction of background in the measurement by the deuterium lamp technique.

Total concentrations of potassium (K) were measured in a flame photometer using a Jenway PFP7 instrument (Jenway Industries PTY LTD, New South Wales, Australia) and specific K filter. Calibration curves were built using standard K solutions with concentrations from 1.0 to 50 mg/L. Usually for this measurement it is necessary to have a dilution of initial sample solutions (1:10) because of high potassium concentrations. Total P concentrations were determined using the vanado-molybdate method [72,75,76]. The measurement was carried out in 880 nm wavelength with a visible-ultraviolet (UV-Vis) double beam spectrophotometer Varian Cary 1E (Varian Inc., Palo Alto, CA, USA).

### 4.4. Statistical Analyses

Data were subjected to over environment two-way analysis of variance (ANOVA), using the mixed model considering genotype treatment effects as fixed effects (variables of interest) and environments as random effect [77]. The Tukey HSD (honestly significant difference) test was used to compare the means at a = 0.05. To explore genotype × environment (G × E) interactions, each location within every year was considered a separate environment, resulting in a total of 8 environments (Table 4). All the analyses were performed using SPSS (version 22) following the experimental design. In addition, a genotype (G) plus genotype × environment (GGE) biplot analysis was used to generate graphs showing the polygon view of the pattern (i.e., which genotype had the highest yield in which environment, or which genotype had the highest value in which trait) using GenStat software (13) [78]. In GGE biplot analysis, genotypes are evaluated for their mean performance and stability and also environmental evaluation and discrimination ability among genotypes in target environments [79,80].

## 5. Conclusions

(i)High environmental effect (62.3–99.8%) for SY, CP and macronutrients of lentil genotypes was observed for both farming systems. The main source of variation for micronutrients was either environment or G × E interaction.(ii)Location Orestiada (Thrace/Northern Greece) showed the highest SY for both organic and conventional farming regardless the year of cultivation and is proposed as the optimum location for lentil cultivation or breeding for high yielding genotypes. Environment 6 (Larissa/Central Greece in 2020) recorded the higher CP values and the highest discriminating ability probably because of the lowest precipitation during the crucial period of pod filling (April–May) in both years. Environments E7 (Thessaloniki/Central Macedonia in 2020) and E8 (Orestiada/Northern Greece in 2020) were the most discriminating environments for producing lentil seeds with high micronutrient contents under both farming systems. These environments had broadly fertile soils and ample soil moisture that may have favored the ability to uptake more micronutrients and could be proposed as ideal for breeding or screening lentil genotypes for micronutrient content.(iii)The genotype effect explained a low percentage of the total variability; however, significant differences among entries revealed two promising genotypes. Cultivar “Samos” (G1) demonstrated a wide adaptation capacity under both organic and conventional farming for SY performance, while the red lentil population “03-24” showed very high level of seed CP, Fe and Mn content regardless of the farming system or the environment cultivated. This genetic material could be further exploited as parental material in breeding programs aiming to develop lentil varieties that could be utilized as “functional” food to cover certain deficiencies via human diet or consist of a significant animal feed ingredient.

## Figures and Tables

**Figure 1 plants-11-03328-f001:**
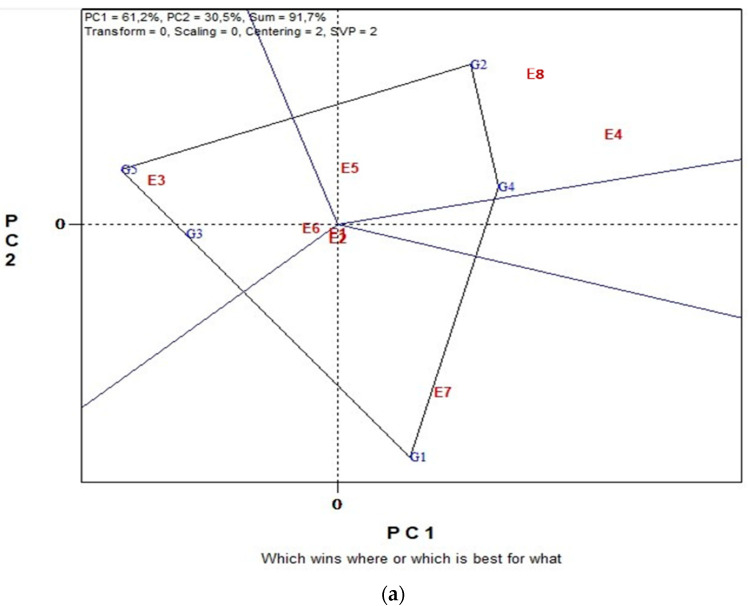
Biplot analysis in organic farming: (**a**) “which-won-where or which is best for what” view of the GGE biplot based on seed yield of five lentil genotypes in eight environments. Environment 1 and Environment 2 coincide; (**b**) “discriminating power vs. representativeness” view of the GGE biplot based on yield of five lentil genotypes in eight environments. Environment 1 and Environment 2 coincide.

**Figure 2 plants-11-03328-f002:**
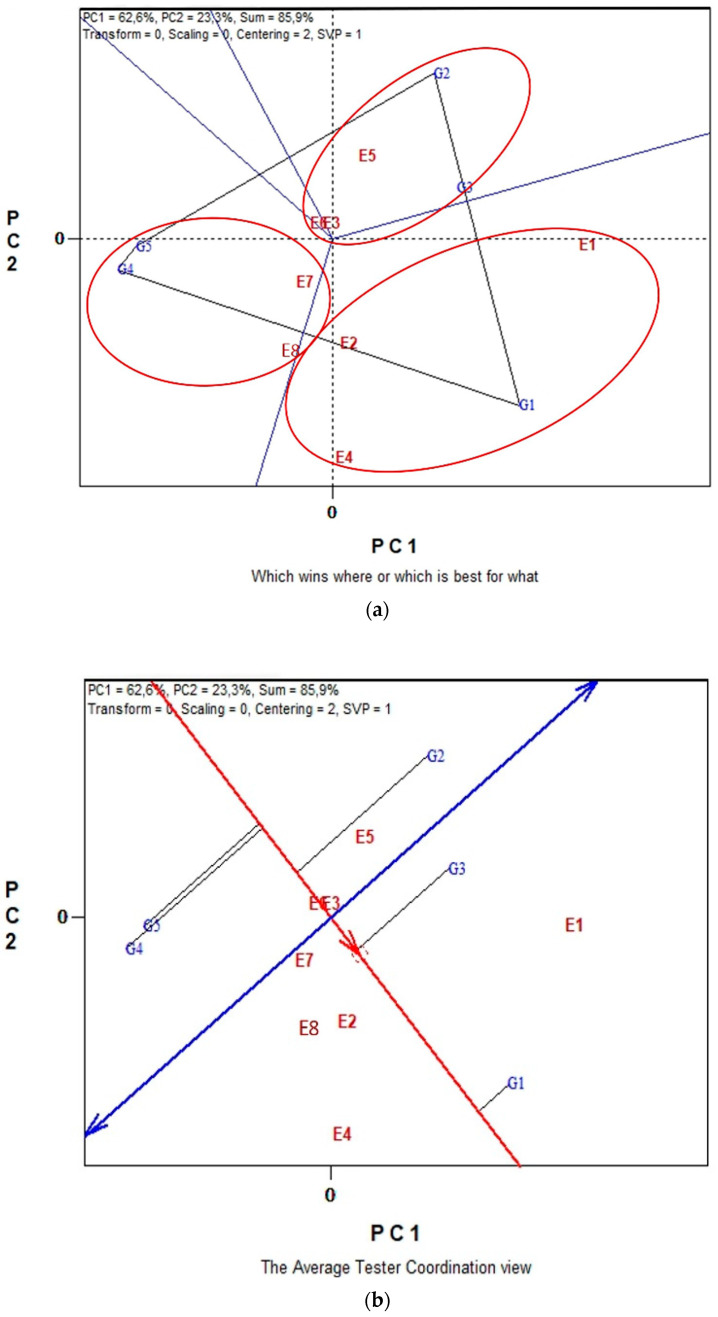
Biplot analysis in conventional farming: (**a**) “which-won-where or which is best for what” view of the GGE biplot based on seed yield of five lentil genotypes in eight environments. Environment 1 and Environment 2 coincide; (**b**) “discriminating power vs. representativeness” view of the GGE biplot based on yield of five lentil genotypes in eight environments. Environment 3 and Environment 4 coincide.

**Figure 3 plants-11-03328-f003:**
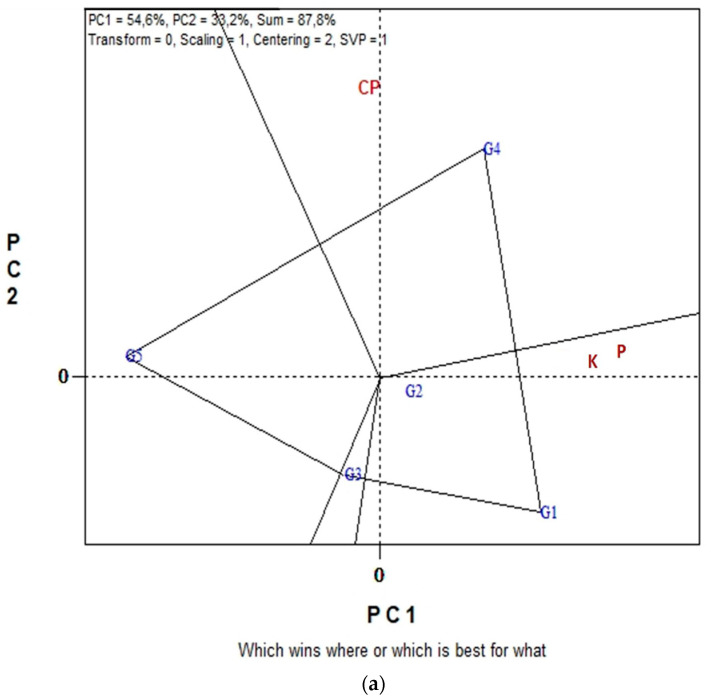
Biplot analysis of seed macronutrient P, K contents and seed crude protein (CP) concentration, in organic farming: (**a**) “which-won-where or which is best for what” view of the GGE biplot based on seed macronutrient P, K contents of five lentil genotypes in eight environments; (**b**) “discriminating power vs. representativeness” view of the GGE biplot based on seed macronutrient P, K contents of five lentil genotypes in eight environments.

**Figure 4 plants-11-03328-f004:**
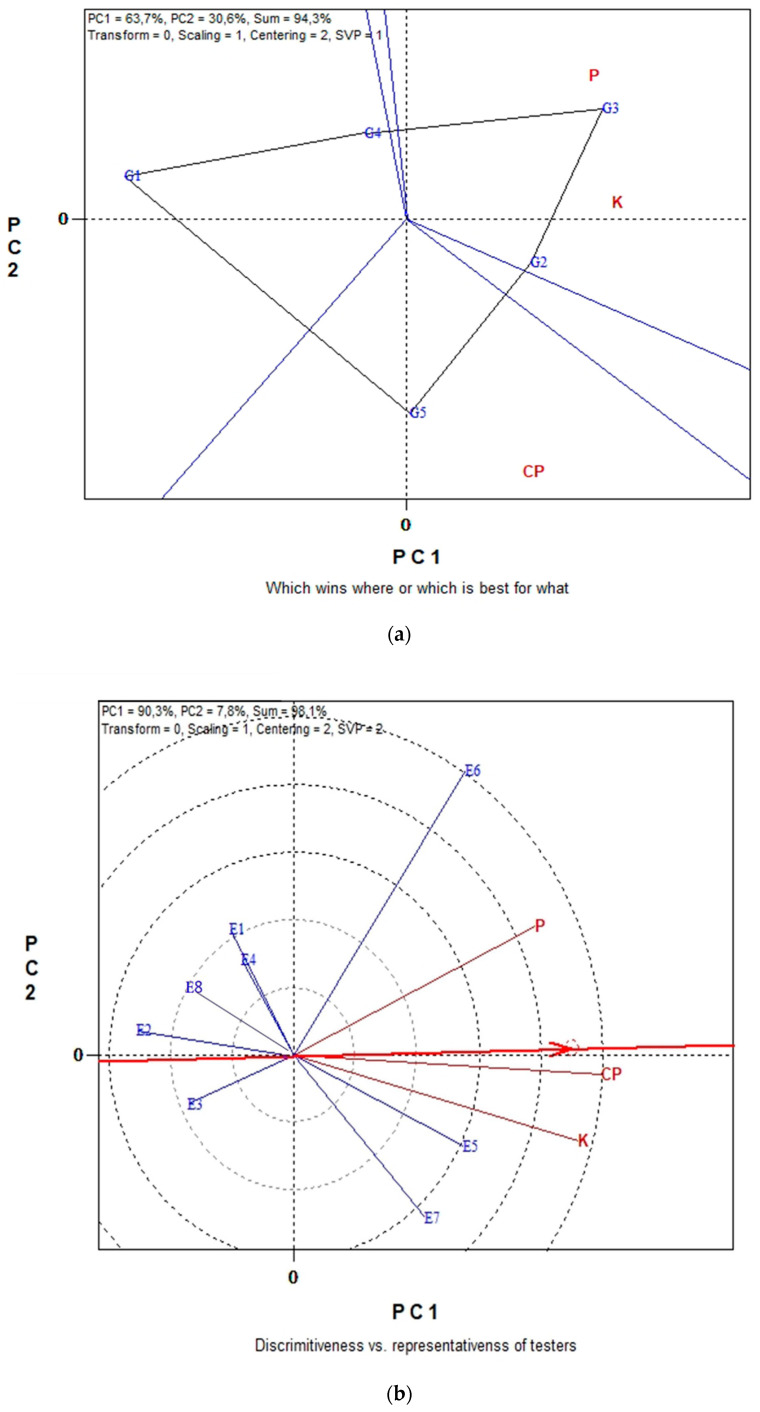
Biplot analysis of seed P, K macronutrient content and seed crude protein (CP) concentration in conventional farming: (**a**) “which-won-where or which is best for what” view of the GGE biplot based on seed macronutrient P, K contents of five lentil genotypes in eight environments; (**b**) “discriminating power vs. representativeness” view of the GGE biplot based on seed macronutrient P, K contents of five lentil genotypes in eight environments.

**Figure 5 plants-11-03328-f005:**
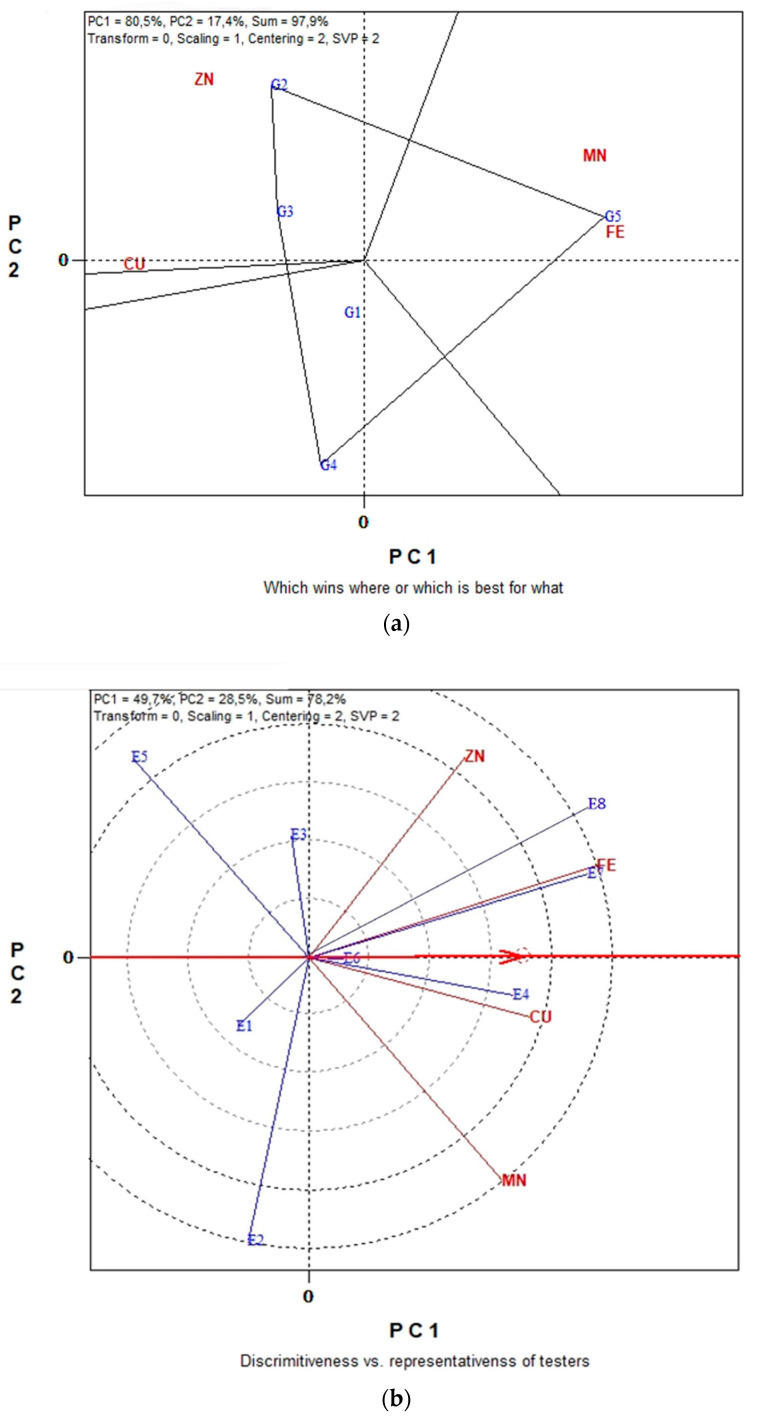
Biplot analysis of seed micronutrient content (Fe, Cu, Mn and Zn) in organic farming: (**a**) “which-won-where or which is best for what” view of the GGE biplot based on seed micronutrient contents (Fe, Cu, Mn and Zn) of five lentil genotypes in eight environments; (**b**) “Discriminating power vs. representativeness” view of the GGE biplot based on seed micronutrient contents (Fe, Cu, Mn and Zn) of five lentil genotypes in eight environments.

**Figure 6 plants-11-03328-f006:**
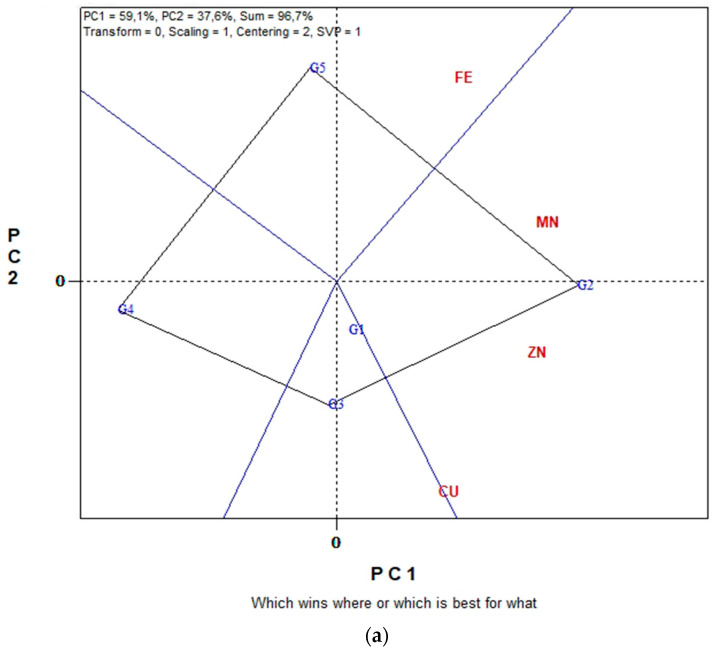
Biplot analysis of seed micronutrient content (Fe, Cu, Mn and Zn) in conventional farming: (**a**) “which-won-where or which is best for what” view of the GGE biplot based on seed micronutrient contents (Fe, Cu, Mn and Zn) of five lentil genotypes in eight environments; (**b**) “discriminating power vs. representativeness” view of the GGE biplot based on seed micronutrient contents (Fe, Cu, Mn and Zn) of five lentil genotypes in eight environments.

**Table 1 plants-11-03328-t001:** Mean squares (MS) analysis of variance (ANOVA) and partitioning of treatment sum of squares (EV%) for seed yield (SY), seed crude protein (% CP), seed macronutrient (% P, K) and micronutrient (Fe, Cu, Mn, Zn, mg kg^−1^) contents of the five lentil genotypes (G) across eight environments (E) in organic and conventional farming.

Organic Farming																	
		SY (t ha^−1^)		CP (%)		P (%)		K (%)		Fe (mg kg^−1^)		Cu (mg kg^−1^)		Mn (mg kg^−1^)		Zn (mg kg^−1^)	
	df	MS	EV%	MS	EV%	MS	EV%	MS	EV%	MS	EV%	MS	EV%	MS	EV%	MS	EV%
**G**	7	0.97 ***	0.77	21.9 ***	4.1	0.1 **	0.26	0.7 ***	5.3	24,063.3 ***	7.7	2.7 ***	7.4	49.4 ***	6.9	19.9 **	1.1
**E**	4	5.63 ***	78.49	277.9 ***	90.1	0.2 ***	97.32	0.5 ***	62.3	63,624.3 ***	35.8	13.68 ***	65.8	105.9 ***	25.9	796.4 ***	72.5
**G × E**	28	0.37 ***	20.74	4.5 ***	5.8	0.1 ***	2.42	0.1 ***	32.4	25,147.3 ***	56.5	1.4 ***	26.8	68.4 ***	67.1	72.5 ***	26.4
**Conventional Farming**																	
**G**	7	6.49 ***	3.76	9.5 ***	1.4	0.1 **	0.53	0.1 ***	0.1	81,720.3 ***	4.1	6.7 ***	17.6	134.9 ***	4.9	168.1 **	8.9
**E**	4	0.53 ***	80.3	356.2 ***	94.5	0.1 ***	92.8	0.3 ***	99.8	61,537.5 ***	53.2	12.21 ***	58	697.5 ***	44.8	468.7 ***	43.5
**G × E**	28	0.32 ***	15.9	0.1 ***	4.1	0.1 ***	6.65	0.1 ***	0.1	12,396.3 ***	42.8	1.3 ***	24.4	195.6 ***	50.3	3.8 ***	47.54

E = (Location × Year); df: degrees of freedom; EV% = % variation with respect to (E + G + G × E) sum of squares; **, *** significant at *p* < 0.01, and *p* < 0.001 levels of probability, respectively.

**Table 2 plants-11-03328-t002:** Comparison of means for seed yield (SY), seed crude protein (% CP), seed macronutrient (% P, K) and micronutrient (Fe, Cu, Mn, Zn, mg kg^−1^) contents of five lentil genotypes (G) at harvest across the eight environments in organic and conventional farming.

Organic Farming								
Genotype	SY (t ha^−1^)	CP (%)	P (%)	K (%)	Fe (mg kg^−1^)	Cu (mg kg^−1^)	Mn (mg kg^−1^)	Zn (mg kg^−1^)
**G1**	1.41 ^b^	23.3 ^a^	0.37 ^c^	1.01 ^b^	114.6 ^b^	8.90 ^b^	15.8 ^b^	47.9 ^ab^
**G2**	1.30 ^a^	24.2 ^b^	0.35 ^ab^	1.02 ^b^	103.9 ^a^	9.00 ^b^	15.5 ^b^	49.4 ^b^
**G3**	1.31 ^a^	23.7 ^a^	0.35 ^ab^	1.00 ^b^	104.8 ^ab^	9.10 ^b^	15.3 ^b^	48.7 ^ab^
**G4**	1.40 ^b^	25.8 ^d^	0.36 ^bc^	1.03 ^b^	106.7 ^ab^	8.84 ^b^	14.3 ^a^	47.5 ^a^
**G5**	1.27 ^a^	24.7 ^c^	0.34 ^a^	0.9 ^a^	177.7 ^c^	8.2 ^a^	18.2 ^c^	47.3 ^a^
**Tukey _HSD (0.05)_**	0.44	0.251	0.116	0.001	161.0	0.140	0.657	4.000
**Conventional Farming**								
**G1**	1.67 ^c^	21.1 ^a^	0.34 ^a^	0.92 ^a^	140.4 ^b^	8.9 ^c^	16.9 ^b^	49.4 ^b^
**G2**	1.48 ^b^	22.4 ^c^	0.37 ^b^	1.01 ^b^	119.8 ^a^	9.3 ^d^	21.0 ^d^	51.5 ^c^
**G3**	1.36 ^a^	21.7 ^b^	0.38 ^b^	1.07 ^c^	141.8 ^b^	9.5 ^d^	16.7 ^b^	47.8 ^b^
**G4**	1.39 ^a^	21.4 ^ab^	0.36 ^ab^	0.99 ^b^	114.9 ^a^	8.6 ^b^	14.5 ^a^	44.81 ^a^
**G5**	1.37 ^a^	22.6 ^c^	0.35 ^a^	1.02 ^b^	256.5 ^c^	8.2 ^a^	17.8 ^c^	46.0 ^a^
**Tukey _HSD (0.05)_**	0.24	0.151	0.063	0.001	309.0	0.83	0.861	4.704

Note: G1 = Samos; G2 = Dimitra; G3 = Elpida; G4 = Thessalia; G5 = 03-24L; different letters within a column indicate significant differences according to Tukey’s HSD (honestly significant difference) test (*p* < 0.05).

**Table 3 plants-11-03328-t003:** Comparison of means for seed yield (SY), seed crude protein (% CP), seed macronutrient (% P, K) and micronutrient (Fe, Cu, Mn, Zn, mg kg^−1^) contents originated from eight environments averaged across five lentil genotypes (G) at harvest in organic and conventional farming.

Organic Farming								
Environment	SY (t ha^−1^)	CP (%)	P (%)	K (%)	Fe (mg kg^−1^)	Cu (mg kg^−1^)	Mn (mg kg^−1^)	Zn (mg kg^−1^)
**E1**	0.96 ^b^	19.7 ^a^	0.38 ^e^	0.95 ^d^	92.3 ^b^	8.9 ^b^	15.3 ^c^	42.6 ^b^
**E2**	1.10 ^bc^	25.7 ^d^	0.28 ^b^	0.90 ^c^	71.9 ^a^	8.7 ^b^	19.1 ^e^	38.0 ^a^
**E3**	1.88 ^d^	21.6 ^b^	0.28 ^b^	0.87 ^bc^	122.7 ^c^	8.8 ^b^	14.1 ^b^	51.9 ^c^
**E4**	1.87 ^d^	19.6 ^a^	0.34 ^c^	0.85 ^b^	155.2 ^d^	9.3 ^c^	20.0 ^e^	57.4 ^d^
**E5**	0.94 ^b^	26.1 ^d^	0.45 ^f^	1.20 ^e^	77.5 ^a^	7.5 ^a^	12.4 ^a^	52.2 ^c^
**E6**	0.46 ^a^	32.0 ^f^	0.49 ^g^	1.10 ^e^	76.7 ^a^	10.1 ^d^	15.3 ^c^	51.6 ^c^
**E7**	1.18 ^c^	27.5 ^e^	0.36 ^d^	1.30 ^f^	266.9 ^e^	9.8 ^d^	16.9 ^d^	52.8 ^c^
**E8**	2.28 ^e^	22.6 ^c^	0.20 ^a^	0.80 ^a^	109.3 ^c^	7.4 ^a^	13.7 ^b^	38.8 ^a^
**Tukey _HSD (0.05)_**	0.44	0.251	0.116	0.001	160.95	0.140	0.657	4.000
**Conventional Farming**								
**E1**	1.66 ^d^	18.5 ^c^	0.38 ^e^	0.94 ^bc^	131.1 ^c^	9.1 ^c^	18.2 ^c^	41.3 ^a^
**E2**	0.89 ^a^	16.0 ^a^	0.28 ^b^	0.87 ^a^	72.9 ^a^	8.2 ^b^	17.3 ^c^	47.4 ^b^
**E3**	1.38 ^c^	16.7 ^b^	0.28 ^b^	0.96 ^c^	136.9 ^c^	8.6 ^b^	13.4 ^b^	51.8 ^c^
**E4**	1.56 ^d^	19.4 ^d^	0.34 ^c^	0.95 ^c^	143.0 ^c^	9.4 ^c^	18.1 ^c^	56.5 ^d^
**E5**	0.91 ^a^	26.8 ^f^	0.45 ^f^	1.20 ^e^	84.0 ^a^	8.5 ^b^	11.7 ^a^	46.9 ^b^
**E6**	1.22 ^b^	27.5 ^g^	0.49 ^g^	1.00 ^d^	71.1 ^a^	9.6 ^c^	13.5	48.5 ^b^
**E7**	0.97 ^a^	27.4 ^g^	0.36 ^d^	1.10 ^e^	487.7 ^d^	10.5 ^d^	29.0 ^d^	51.5 ^c^
**E8**	2.89 ^d^	22.4 ^e^	0.20 ^a^	0.9 ^ab^	110.7 ^b^	7.4 ^a^	13.7 ^b^	39.4 ^a^
**Tukey _HSD (0.05)_**	0.24	0.151	0.063	0.001	309.037	0.83	0.861	4.704

Note: E1: Domokos 2019; E2: Larissa 2019; E3: Thessaloniki 2019; E4: Orestiada 2019; E5: Domokos 2020; E6: Larissa 2020; E7: Thessaloniki 2020; E8: Orestiada 2020; different letters within a column indicate significant differences according to Tukey’s HSD (honestly significant difference) test (*p* < 0.05).

**Table 4 plants-11-03328-t004:** Meteorological data, soil order and basic soil parameters of the 8 Environments (Location × year) over 2019 and 2020 growing seasons. Soil parameters refer to conventional (Conv.) and organic (Org.) farming conditions at 0–30 cm depth.

Environment	Location	Year	PrS ^1^ (mm)	PrA-M ^2^ (mm)	T ^3^ (°C)	Soil Order ^4^	Soil Texture		pH _(1:1)_		EC ^5^		SOM ^6^ %		CaCO_3_ %		P_Olsen_ mg kg^−1^	
							Conv.	Org.	Conv.	Org.	Conv.	Org.	Conv.	Org.	Conv.	Org.	Conv.	Org.
E1	Domokos	2019	566.1	81.8	11.3	*Vertisols*	C	C	7.1	7.1	0.28	0.34	1.7	1.4	2.0	1.5	18.0	26.0
E2	Larissa	2019	479.2	72.0	13.9	*Fluvisols*	C	C	7.7	7.9	0.43	0.52	1.1	1.2	1.0	1.7	3.6	7.3
E3	Thessaloniki	2019	399.6	107.8	14.0	*Fluvisols*	CL	CL	8.1	8.0	0.62	0.69	1.0	1.3	5.0	4.5	11.0	6.7
E4	Orestiada	2019	367.2	118.3	12.6	*Fluvisols*	C	C	7.8	7.9	0.41	0.52	1.7	1.8	1.0	1.5	7.3	10.2
E5	Domokos	2020	684.7	186.4	11.5	*Vertisols*	SCL	CL	7.1	6.9	0.40	0.45	1.6	1.4	2.0	1.5	45.0	31.0
E6	Larissa	2020	453.2	99.0	14.4	*Fluvisols*	C	C	7.1	7.2	0.44	0.50	1.2	1.0	2.0	2.0	19.0	7.6
E7	Thessaloniki	2020	524.8	168.4	14.4	*Fluvisols*	CL	CL	8.1	-	0.62	-	1.0	-	5.0	-	11.0	-
E8	Orestiada	2020	432.6	182.8	12.5	*Fluvisols*	SiCL	SiCL	7.3	7.2	0.73	0.72	1.6	2.0	2.0	2.0	8.4	7.3

Note: PrS ^1^ = Precipitation during the growing season (November to July), ^2^ PrA-M = Precip. April–May (Depending on climate zone of the location this period mostly represents the beginning of flowering to pod filling. This crucial stage may be differentiated accordingly in some locations), ^3^ Season Avg. T (°C) = Average temperature in the growing season (November to July), ^4^ Prevailing soil classification (soil order) [48,49]. ^5^ EC = Electrical Conductivity (mS cm^−1^) (25 °C), ^6^ SOM = Soil organic matter.

## Data Availability

The data presented in this study are available on request from the corresponding author.

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
