# Peer review of "Seed Yield, Crude Protein and Mineral Nutrients of Lentil Genotypes Evaluated across Diverse Environments under Organic and Conventional Farming"

_plants, 2022, doi:10.3390/plants11233328_

Round 1

Reviewer 1 Report

Tziouvalekas et al. presented a practical analysis of the results of agricultural experiments. The authors analyzed variations between genotypes (G), environments (E) and genotypes by environment (GE interaction) using GGE biplot analysis and ANOVA. In my opinion, there was no use of other statistical parameters (parametric or non-parametric analyzes) for the GEI study, which would allow assigning appropriate ranks to genotypes and environments. Also, the biological approach to discussing the GEI was missing from the discussion itself. My comment does not question the validity of the methods used by the authors. The title proposed by the authors is very long but describes the entire article comprehensively. In addition, I would like to point out:

- in lines 173 and 180 there is p ≤ 0.05, which needs to be changed to p < 0.05.

- in Figures 1-6 the font size is too small for the graphs.

- in the discussion is too much of subsections, I would recommend changing it as follows:

2.1. Combined variance analysis for yield and Inorganic Compounds of Lentil Genotypes

2.2. GGE biplot analysis for Seed Yield in Organic and Conventional farming

2.3. GGE biplot analysis for Seed Crude Protein and Macro-nutrient in Organic and Conventional farming

2.4 GGE biplot analysis for Seed Micro-nutrient in Organic and Conventional farming

I recommend this article for minor revision.

Author Response

We thank the reviewer for the positive comments on our study and for the interesting information, useful for improving it.

Responses:

- in lines 173 and 180 there is p ≤ 0.05, which needs to be changed to p < 0.05.

Response: We revised according to the reviewer's request. In lines 173 and 180 probability level changed to p < 0.05, as well as in lines 142, 203 of the initial pdf file.

- in Figures 1-6 the font size is too small for the graphs.

Response: We changed according to the reviewer's request.

- in the discussion is too much of subsections, I would recommend changing it as follows:

2.1. Combined variance analysis for yield and Inorganic Compounds of Lentil Genotypes

2.2. GGE biplot analysis for Seed Yield in Organic and Conventional farming

2.3. GGE biplot analysis for Seed Crude Protein and Macro-nutrient in Organic and Conventional farming

2.4 GGE biplot analysis for Seed Micro-nutrient in Organic and Conventional farming

Response: We assume the reviewer is referring to “Results” section. Subsection titles were modified according to reviewer’s suggestions

Reviewer 2 Report

The manuscript deals with the comparison of agronomic and nutritionel traits of five gentypes of lentil cuntivated under two contrasted agricultural modes during two successive years.

The topic of the manuscript is of great interst and meets the expectations of Plants.

The manuscript, however needs to be modified.

1- why authors did not chosen a split plot design due to two factors studied. This is more judicious since, as expected the effect of agricultural modes was predominant comparing to genotype effect.

2-Plant nutrition for organic cultivation was not presented

3-point iii of the conclusion should be rephrased not in the line of the objectives of the study.

Author Response

We thank the reviewer for the positive comments on our study and for the interesting information, useful for improving it.

Responses

1- why authors did not chosen a split plot design due to two factors studied. This is more judicious since, as expected the effect of agricultural modes was predominant comparing to genotype effect.

Response: The experiment has been performed and analyzed according to RCB design as the main purpose was to identify the most suitable genotypes and the ideal environment within each farming system (i.e., conventional and organic). Farming system was not a factor in this study. This is clear from the presentation of the results where all results are presented separately for each farming system and from the discussion where no comparisons are made between the two farming systems.

2-Plant nutrition for organic cultivation was not presented

Response: The respective information was added in the text. (Last paragraph in section “4.2 Locations and growing conditions” in M&M, highlighted yellow, lines 621-626 in the revised MS)

3-point iii of the conclusion should be rephrased not in the line of the objectives of the study.

Response: The objectives were rephrased to be in agreement with the point iii in the conclusions paragraph

Round 2

Reviewer 2 Report

Dear  Authors,

The manuscript was improved and many concerns in this version were addressed.

I am still not convinced about not taking the farming systems into account as a factor in the analyses of variance.
This is all the more important as the grain yield (Table 3) is different for the same genotypes between the farming systems.
I suggest that these results be further highlighted in the discussion by explaining the sources of this interaction (genotype X farming system) without revisiting the ANOVA.

Author Response

We thank the reviewer for his valuable comment. We totally agree that the farming system had a profound effect on the performance of genotypes on SY and most of the traits studied.

As we have indicated in our first response with regard to this issue in this study the main objective was the performance of the genotypes in each environment (E was appended as Location×Year). We have chosen two-way ANOVA considering genotype treatment effects as fixed effects and environments (E) as random effect. Genotypes was the fixed effect as was the variable of interest. It was out of our scope to include the farming system as a separate factor and thus the experiment was not designed as a split-plot, but as two separate RCB designs.

However, we added a short paragraph (lines 385-388, highlighted with yellow) in the discussion session highlighting the importance of the farming system effect.